# State Parameter Estimation of Intelligent Vehicles Based on an Adaptive Unscented Kalman Filter

Yu Wang [1], Yushan Li [2] and Ziliang Zhao [2,*]

1 College of Mechanical and Electronic Engineering, Shandong University of Science and Technology, Qingdao 266590, China
2 College of Transportation, Shandong University of Science and Technology, Qingdao 266590, China
* Correspondence: zhaoziliang1@sdust.edu.cn

**Abstract:** The premise of vehicle intelligent decision making is to obtain vehicle motion state parameters accurately and in real-time. Several state parameters cannot be measured directly by vehicle sensors, so estimation algorithms based on filtering are effective solutions. The most representative algorithm is the Kalman filter, especially the standard unscented Kalman filter (UKF) that has been widely used in vehicle state estimation because of its superiority in dealing with nonlinear filtering problems. However, although the UKF assumes that the noise statistics of the system are known, due to the complex and changeable operating conditions, sensor aging and other factors, these noises vary. In order to realize high-precision vehicle state estimation, a noise-adaptive UKF algorithm is proposed in this article. The maximum a posteriori (MAP) algorithm is used to dynamically update the noise of the vehicle system, and it is embedded into the update step of the UKF to form an adaptive unscented Kalman filter (AUKF). The system will dynamically update the noise when noise statistics are unknown and prevent filter divergence by adjusting the mean and covariance of the estimated noise to improve accuracy. On this basis, the proposed method is verified by the joint simulation of CarSim and Matlab/Simulink, confirming that the AUKF performs better than the standard UKF in estimation accuracy and stability under different degrees of noise disturbance, and the estimation accuracy for the yaw rate, side slip angle and longitudinal velocity is improved by 20.08%, 40.98% and 89.91%, respectively.

**Keywords:** adaptive unscented Kalman filter; noise statistic estimator; vehicle state parameter estimation





## 1. Introduction

Obtaining the relevant parameters of intelligent vehicles' driving state accurately and in real-time is the premise of active intelligent vehicle safety control. Because of the high cost and technical constraints of some intelligent vehicle state parameter sensors, this information cannot be measured directly. In the hopes of obtaining vehicle critical state information in a more economically feasible manner, intelligent vehicle state parameter estimation based on low-cost vehicle sensors and related algorithms has become a research hotspot [1–3]. State parameters characterizing vehicle stability in intelligent vehicle active safety control systems have become a key focus of related research [4]. Due to the complex and changeable working conditions of the environment, sensor aging and changeable noise in the actual driving process, estimation divergence often occurs, which leads to a reduction in estimation accuracy.

Estimation methods using nonlinear observers and the Kalman filter (KF) have received extensive attention by scholars. In [5,6], nonlinear observers were used to estimate vehicle states. Although these methods were proven to be effective under some conditions, the accurate acquisition of vehicle model parameters had a great influence on the estimation accuracy. Traditional Kalman filtering algorithms, such as the extended Kalman filter (EKF) and unscented Kalman filter (UKF), are implemented by recursive iteration for nonlinear

systems, which are simple to calculate and easy to implement, and obtain better estimation results. Therefore, the KF algorithm has become one of the most widely used algorithms in research [7,8].

Some scholars have studied the vehicle state estimation based on EKF [9–11]. Compared with the EKF algorithm, the UKF algorithm abandons the Jacobian matrix for solving nonlinear functions, which reduces the amount of calculations and improves the accuracy and stability [12]. Liu Yingjie et al. [13] combined the UKF with genetic particle swarm optimization (PSO) to reduce computational complexity, and optimized the convergence speed and the estimation accuracy of vehicle state parameters. The UKF algorithm utilizes a noise covariance matrix to describe the process noise caused by model uncertainty and the measurement noise superimposed by the sensor error in the measurement process. However, these noises are generated randomly and not fixed in practice.

For this reason, scholars have proposed an adaptive adjustment mechanism of the noise covariance matrix and developed an adaptive Kalman filtering method. For example, Shen Fapeng et al. [14] made use of the ability of the particle filter algorithm to solve nonlinear and non-Gaussian problems, combined with the iterative extended Kalman algorithm for vehicle state estimation, and obtained high estimation accuracy. Li Gang [15] improved the estimation accuracy by improving the adaptive rules on the basis of the Sage–Husa adaptive EKF algorithm. BOADA et al. [16] first estimated the cornering angle with the help of an adaptive neuro-fuzzy inference system, took the estimated value as the measurement variable of the UKF and obtained an accurate cornering angle by minimizing the variance of the estimated mean square error. Wang Zhenpo et al. [17] combined fuzzy control with the unscented Kalman filter algorithm to realize the adaptive adjustment of the system measurement noise covariance matrix. Li Jiabo et al. [18] carried out joint modeling of the improved least squares support vector machine (LSSVM) and adaptive UKF to control the estimation error of SOC within 2%. Xue Zhongjin et al. [19] used unscented transform and statistical linearization to suppress outliers. On this basis, an iterative weighted least squares method based on M-estimation is used to deal with process uncertainty, innovation and observation outliers, which improves the robustness of the estimation process. Wang Yan et al. [20] proposed an embedded cubic Kalman filtering algorithm based on the coupled vehicle model to ensure the accuracy of preceding vehicles (PVS) state estimation while reducing the communication rate when the communication bandwidth is limited. When the communication rate is reduced to 37.55%, the estimation accuracy is still higher than that achieved with the cubic Kalman filter. In addition, many scholars have also carried out state estimation research based on cubature Kalman filter (CKF) [21–24], and have achieved good estimation results.

Through the analysis of the existing research results, ensuring the stability of the estimation method and avoiding estimation divergence while improving the estimation accuracy when an adaptive adjustment mechanism is introduced is a key problem [25]. Most of the current research is to set the covariance matrix of the observation noise to a fixed value and then make dynamic adjustments. This method is improved compared with the previous method, but in actual engineering applications, the process noise and measurement noise are dynamically changing, and thus online estimation and identification represent an improved method that can adapt to the real conditions. Therefore, we propose a noise adaptive UKF algorithm to obtain vehicle state parameters accurately in the presence of noise interference.

The core contributions of this study are as follows: (1) Using the maximum a posterior (MAP) algorithm to dynamically update the noise of vehicle system; (2) improve the noise statistic estimator so that the estimated noise covariance is positive and kept within a certain regular range; and (3) to have the above improved noise statistical estimation method embedded into the update step of the UKF to form an adaptive unscented Kalman filter (AUKF) algorithm, which can prevent filter divergence by adjusting the mean and covariance of measurement noise and the estimated noise to improve accuracy. In the simulation process, we set the process noise and observation noise in different time periods

to different values. The results confirm that the estimation accuracy and stability of the AUKF are better than standard UKF under different degrees of noise disturbance.

The chapters of this article are arranged as follows:

In Section 2, we define the vehicle coordinate system and establish the vehicle dynamics model. We introduce the architecture of adaptive untraced Kalman filter (AUKF) in Section 3, and conduct a simulation comparison analysis of AUKF and standard UKF in Section 4. Finally, Section 5 summarizes the conclusion of the article.

## 2. Vehicle Dynamics Model

Considering the complexity of modeling and the need for real-time calculations, we introduced the longitudinal motion degree of freedom into the two-degrees-of-freedom (2-DOF) vehicle model [26] to form a 3-DOF model [27] with yaw displacement, lateral displacement and longitudinal displacement. The model diagram is shown in Figure 1. We assume that the vehicle is symmetrical to the X-axis and take the center of mass as the origin to establish the XOY coordinate system.

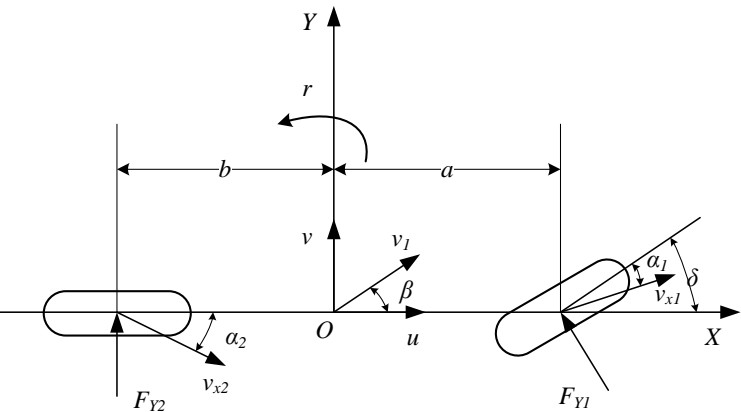

**Figure 1.** 3-DOF vehicle model.

u and t are the longitudinal speed and the lateral speed; $F_{Y1}$ and $F_{Y2}$ are the lateral forces of the front and rear axles; $v_1$ is the centroid velocity; $\alpha_1$ and $\alpha_2$ are the side deflection angle; $v_1$ is the centroid velocity; and $v_{x1}$ and $v_{x2}$ are the speed of the midpoint of the front and rear axles of the vehicle.

The motion equation of the vehicle includes two input variables, three state variables and one measurement variable:

State equation:

$$
\begin{cases}
\dot{r} = \dfrac{a^2 k_1 + b^2 k_2}{I_x v_x} r + \dfrac{a k_1 - b k_2}{I_z} \beta - \dfrac{a k_1}{I_z} \delta \\[2mm]
\dot{\beta} = \left( \dfrac{a k_1 - b k_2}{m v_x^2} - 1 \right) r + \dfrac{k_1 + k_2}{m v_x} \beta - \dfrac{k_1}{m v_x} \delta \\[2mm]
\dot{v}_x = r \beta v_x + a_x
\end{cases}
\tag{1}
$$

Measurement equation:

$$
a_y = \frac{a k_1 - b k_2}{m v_x} r + \frac{k_1 + k_2}{m} \beta - \frac{k_1}{m} \delta
\tag{2}
$$

In the formula, $\beta$ and $r$ are the center of mass angle and the yaw angular velocity; $v_x$ is the longitudinal speed; $a_x$ is the longitudinal acceleration; $a_y$ is the lateral acceleration; $a$ and $b$ are the distance from the center of mass to the front and rear axles; $k_1$ and $k_2$ are the equivalent lateral cornering of the front and rear axle; $I_z$ is the moment of inertia around the Z axis; $\delta$ is the front wheel angle; and $m$ is the vehicle mass.

### 3. AUKF Algorithm

In order to facilitate state estimation, we use the following formula to express the state–space equation:

$$X_{k+1} = f(X_k, k) + \omega_k \tag{3}$$

$$Z_{k+1} = h(X_{k+1}, k+1) + v_{k+1} \tag{4}$$

where $X_k$ and $Z_k$ are the state vector and output vector; $\omega_k$ and $v_k$ are the system excitation noise and measurement noise; $q_k$ and $r_k$ are the mean value of $\omega_k$ and $v_k$; and $Q_k$ and $R_k$ are covariance matrix of $\omega_k$ and $v_k$.

Based on the traditional UKF iterative framework [28,29], AUKF includes the following two steps:

(1) The UKF obtains the sigma point using the following formula:

$$\begin{cases} \chi_0 = \overline{X}_k & i = 0 \\ \chi_{i,k} = \overline{X}_k + \left( \sqrt{(n+\lambda)P_k} \right)_i & i = 1, \cdots, n \\ \chi_{i,k+1-q} = \overline{X}_k - \left( \sqrt{(n+\lambda)P_k} \right)_i & i = L+1, \cdots 2n \end{cases} \tag{5}$$

where $\chi_{i,k}$ are the sigma points, $n$ is dimension of the state vector, $P_k$ is the system state error matrix, and $\lambda$ is the scale factor.

The next step in the UKF process involves making a one-step prediction for each sigma point using the system equations. The predicted sigma point is obtained as

$$X_{i,k+1|k} = f(\chi_{i,k}, k) + q_k \tag{6}$$

Then, the predicted values of the system state variables $\overline{X}_{k+1|k}$ and covariance matrix $P_{k+1|k}$ are obtained as

$$\overline{X}_{k+1|k} = \sum_{i=0}^{2n} W_i^{(m)} X_{i,k+1|k} \tag{7}$$

$$P_{k+1|k} = \sum_{i=0}^{2n} W_i^{(c)} \left[ X_{i,k+1|k} - \overline{X}_{k+1|k} \right] \left[ X_{i,k+1|k} - \overline{X}_{k+1|k} \right]^T + Q_{k+1}^T \tag{8}$$

where $\begin{cases} W_0^{(m)} = W_0^{(c)} = \lambda/(n+\lambda), i = 0 \\ W_i^{(m)} = W_i^{(c)} = 1/2(n+\lambda), i = 1, \cdots 2n \end{cases}$ . Subsequently, the observed predicted values of sigma points are calculated, and the covariance matrix of observed variable $P_{yy}$ is obtained via weighted summation,

$$Y_{i,k+1|k} = h\left( X_{i,k+1|k}, k+1 \right) + r_{k+1} \tag{9}$$

$$\overline{y}_{k+1|k} = \sum_{i=0}^{2n} W_i^{(m)} Y_{i,k+1|k} + r_{k+1} \tag{10}$$

$$P_{yy} = \sum_{i=0}^{2n} W_i^{(c)} \left[ Y_{i,k+1|k} - \overline{y}_{k+1|k} \right] \left[ Y_{i,k+1|k} - \overline{y}_{k+1|k} \right]^T + R_{k+1} \tag{11}$$

In addition, the covariance matrix $P_{xy}$ between $\overline{X}_{k|k-1}$ and $\overline{y}_{k|k-1}$ are obtained as

$$P_{xy} = \sum_{i=0}^{2n} W_i^{(c)} \left[ X_{i,k+1|k} - \overline{X}_{k+1|k} \right] \left[ Y_{i,k+1|k} - \overline{y}_{k+1|k} \right]^T \tag{12}$$

Finally, the gain matrix $K_k$ is calculated and the state variable $\overline{X}_k$ and error covariance matrix $P_k$ are updated.

$$K_{k+1} = P_{xy} P_{yy}^{-1} \tag{13}$$

$$\overline{X}_{k+1} = \overline{X}_{k+1|k} + K_{k+1}\left(Z_{k+1} - \overline{y}_{k+1|k}\right) \tag{14}$$

$$P_{k+1} = P_{k+1|k} - K_{k+1}P_{yy}K_{k+1}^{\mathrm{T}} \tag{15}$$

(2) When the measurement noise and process noise are fixed, the UKF algorithm works normally and can complete the estimation of vehicle state parameters. However, the process noise and observation noise are generated randomly in practice. To solve this problem, a noise statistical estimator is designed using the MAP algorithm [30], and a MAP-based AUKF algorithm theory is proposed. The noise update steps are as follows:

$$\hat{r}_{k+1} = (1 - d_{k+1})\hat{r}_k + d_{k+1}\left[Z_{k+1} - \sum_{i=0}^{2n} W_i^m h_{k+1}\left(X_{i,k+1|k}\right)\right] \tag{16}$$

$$\hat{R}_{k+1} = (1 - d_{k+1})\hat{R}_k + d_{k+1}\left[\varepsilon_{k+1}\varepsilon_{k+1}^{\mathrm{T}} - \sum_{i=0}^{2n} W_i^{(c)}\left(Y_{i,k+1|k} - \overline{y}_{k+1|k}\right)\left(Y_{i,k+1|k} - \overline{y}_{k+1|k}\right)^{\mathrm{T}}\right] \tag{17}$$

$$\hat{q}_{k+1} = (1 - d_{k+1})\hat{q}_k + d_{k+1}\left[\overline{X}_{k+1} - \sum_{i=0}^{2n} W_i^{(m)} f\left(\chi_{i,k+1|k}\right)\right] \tag{18}$$

$$\hat{Q}_{k+1} = (1 - d_{k+1})\hat{Q}_k + d_{k+1}\left[K_{k+1}\varepsilon_{k+1}\varepsilon_{k+1}^{\mathrm{T}}K_{k+1}^T + P_{k+1} - \sum_{i=0}^{2n} W_i^{(c)}\left(X_{i,k+1|k} - \overline{X}_{k+1|k}\right)\left(X_{i,k+1|k} - \overline{X}_{k+1|k}\right)^{\mathrm{T}}\right] \tag{19}$$

where $\varepsilon_{k+1} = Z_{k+1} - h\left(\overline{X}_{k+1|k}, k+1\right) - r_{k+1}$, $d_{k+1} = (1 - b)/\left(1 - b^{k+1}\right)$, and $0 < \mathrm{b} < 1$ is the forgetting factor. In general, the filter cooperates well with the conventional algorithms (5)~(19). However, there is subtraction in Equations (17) and (19) that can produce negative $\hat{R}_{k+1}$ and $\hat{Q}_{k+1}$ matrices. Therefore, we make the following improvements to the noise statistical estimator to avoid this kind of situation:

1. Calculate the $\hat{R}_{k+1}$ using Equation (17); if $\hat{R}_{k+1} < 0$, then:

$$R_{k+1} = \hat{R}_{k+1} + d_{k+1}(\sum_{i=0}^{2n} W_i^{(c)}\left[Y_{i,k+1|k} - \overline{y}_{k+1|k}\right]\left[Y_{i,k+1|k} - \overline{y}_{k+1|k}\right]^T) \tag{20}$$

2. Calculate the $\hat{Q}_{k+1}$ using Equation (19); if $\hat{Q}_{k+1} < 0$, then:

$$Q_k = \hat{Q}_{k+1} + d_{k+1}(\sum_{i=0}^{2n} W_i^{(c)}\left[X_{i,k+1|k} - \overline{X}_{k+1|k}\right]\left[X_{i,k+1|k} - \overline{X}_{k+1|k}\right]^T) \tag{21}$$

Therefore, if $R_0$ and $Q_0$ are positive definite matrices, $R_k$ and $Q_k$ can be positive definite matrices with any given $k$.

Figure 2 is the frame diagram of the estimation process of the AUKF algorithm. The specific iterative process is as follows.

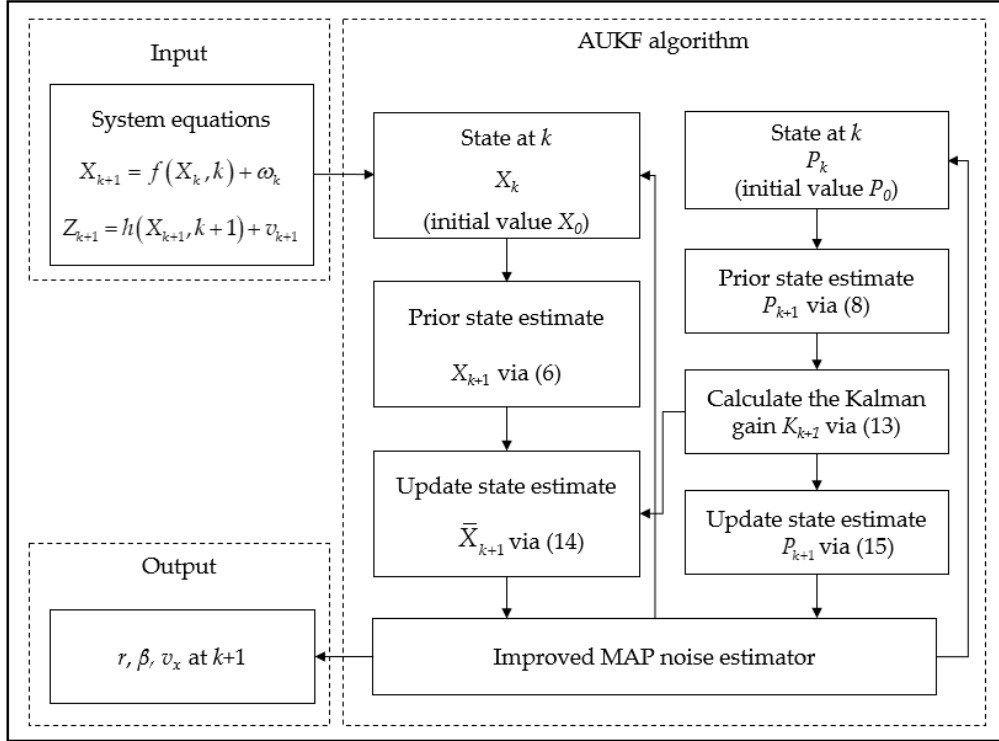

**Figure 2.** The framework of the AUKF algorithm.

## 4. Simulation Results and Analyses

According to the literature [31,32], combining the CarSim and MATLAB/Simulink simulation platform can effectively verify the estimation algorithm. The control quantity and observation output of the vehicle are input into the UKF algorithm model, and the three state variables are estimated in real-time. We compare the estimated results of UKF and AUKF with the ideal values of CarSim output, and obtain the maximum estimation error and the percentage improvement of estimation accuracy, so as to verify the effectiveness of the AUKF algorithm. The parameters of the vehicle model used in this paper are given: $m = 1310$ kg, $a = 1.015$ m, $b = 1.895$ m and $I_z = 1536.7$ kg·m$^2$.

The UKF algorithm and AUKF algorithm are compared at different speeds under double lane change and serpentine conditions. The friction coefficient between tire and road surface is 0.85, and the sampling time is Ts = 0.01 s. In order to highlight that the proposed method can cope with different degrees of noise disturbance, we set the variance of the noise matrix of the square normal condition analysis in the first half to 0.001, and increase the variance of the noise matrix in the second half tenfold, so as to show that the vehicle can still achieve adaptive filtering under different noise levels. The estimation accuracy of the algorithm usually chooses the root mean square error (RMSE) to describe:

$$RMSE = \sqrt{\frac{1}{M}\sum_{k=1}^{M}(x_k - \hat{x}_k)^2} \tag{22}$$

where $M$ represents the total time step of the run, and $k$ represents the time step of one run.

### 4.1. Simulation Analysis of Double Lane Change Condition

(1) We fixed the vehicle speed at 40 km/h and initialized the state vector as $x_0 = [0,0,40/3.6]$. Figures 3a, 4a and 5a show the simulation results of UKF, AUKF and ideal values.

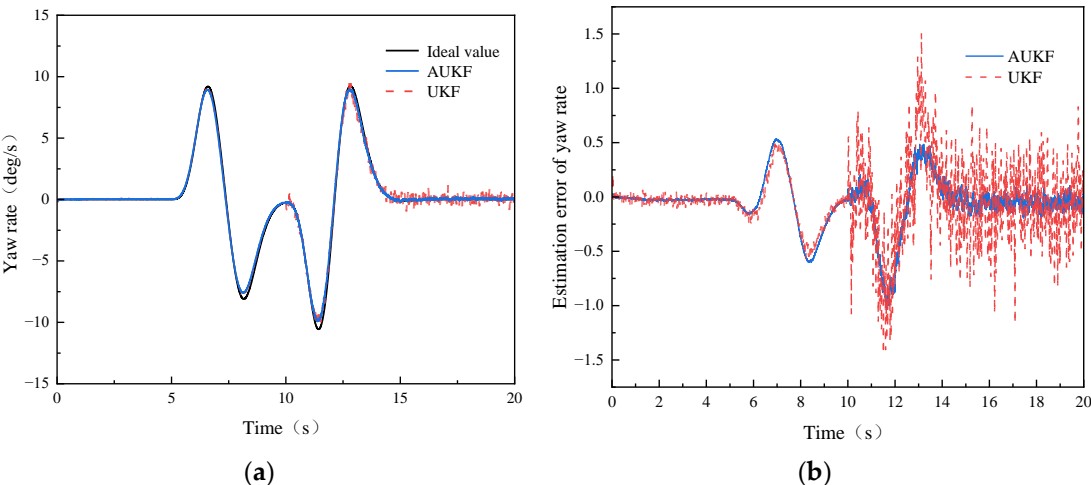

**Figure 3.** Simulation results of yaw rate. (**a**) Comparison of estimated results and (**b**) comparison of estimated error.

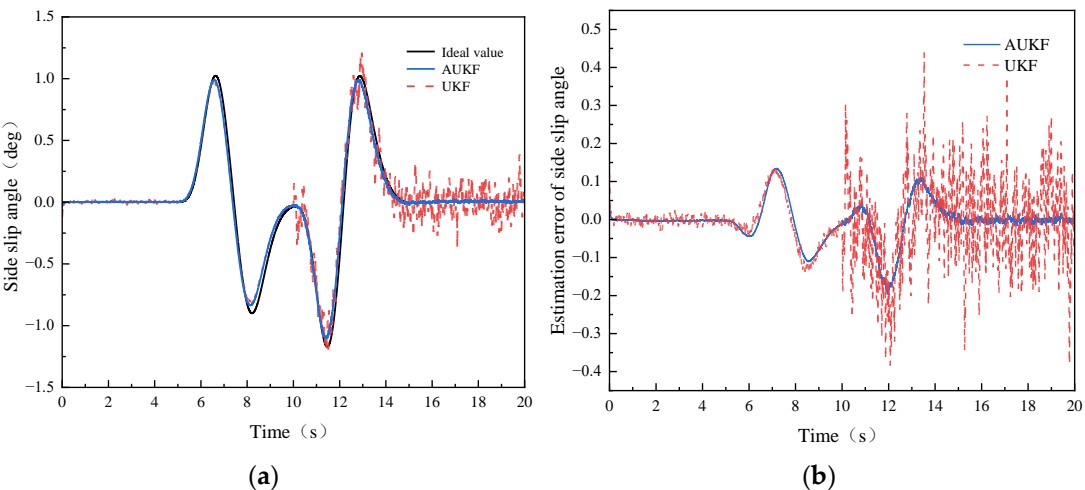

**Figure 4.** Simulation results of side slip angle. (**a**) Comparison of estimated results and (**b**) comparison of estimated error.

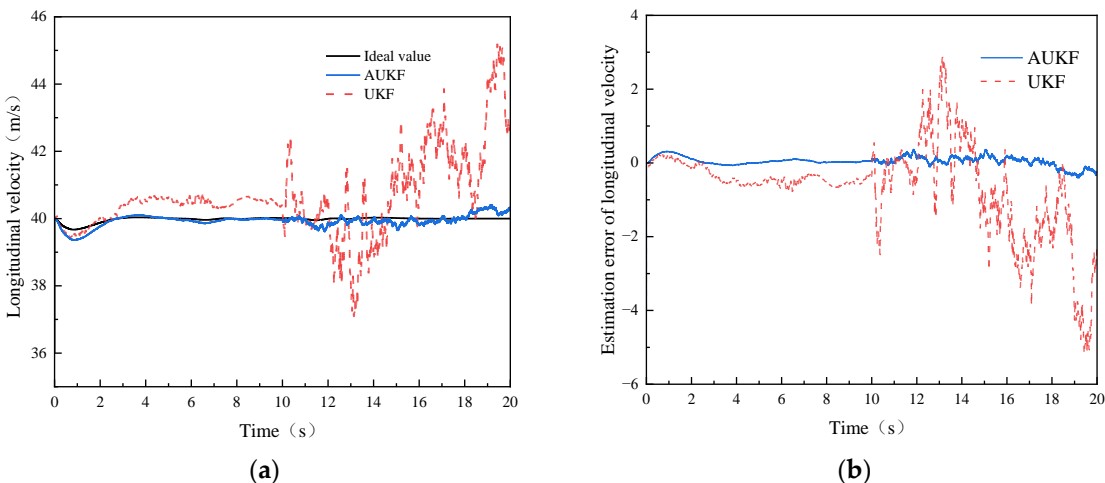

**Figure 5.** Simulation results of longitudinal velocity. (**a**) Comparison of estimated results and (**b**) comparison of estimated error.

In Figures 3a and 4a, we can see that the vehicle changes lanes at 5–15 s. At the inflection point of the curve, the standard UKF deviates from the reference value, while the AUKF can track the ideal value effectively. At the same time, due to the change in process noise after 10 s, UKF diverges in the estimation, which is larger than that in the previous 10 s, indicating that UKF algorithm cannot accurately estimate the corresponding state parameters when there are different degrees of noise disturbance. The estimation results of the AUKF algorithm are largely consistent with the ideal values in the whole working condition, and the effect of the AUKF algorithm is ideal. The maximum instantaneous error of the yaw rate and side slip angle is 1.034 deg/s and 0.195 deg.

As shown in Figure 5a, the UKF algorithm has a divergence in the estimation, especially after the variance of the noise matrix increases tenfold, which shows that the UKF algorithm cannot accurately estimate the corresponding state parameters when there are different degrees of noise disturbance. The AUKF in the first 10 s is largely consistent with the ideal value, and the error is within an acceptable range in the last 10 s. The effect of the AUKF algorithm is ideal and the maximum instantaneous error of the longitudinal velocity is 0.405 m/s.

The estimation errors of each algorithm are shown in Figures 3b, 4b and 5b. The RMSE of the UKF and AUKF are given in Table 1.

**Table 1.** RMSE of the two algorithms during the entire duration of the process.

| Estimation Algorithm | RMSE of State Variable Estimation | | |
|---|---|---|---|
| | Yaw Rate | Side Slip Angle | Longitudinal Velocity |
| UKF | 0.3188 | 0.0898 | 1.3941 |
| AUKF | 0.2502 | 0.0530 | 0.1364 |

As can be seen from Table 1, in the estimation of the three kinds of vehicle states, the RMSE estimated by the AUKF algorithm is the smallest, demonstrating its higher estimation accuracy. The estimation accuracy for yaw rate, side slip angle and longitudinal velocity was improved by 21.52%, 40.98% and 90.22%, respectively.

(2) We fixed the vehicle speed at 80 km/h and initialize the state vector as $x_0 = [0,0,80/3.6]$. Figures 6a, 7a and 8a show the simulation results of UKF, AUKF and ideal values.

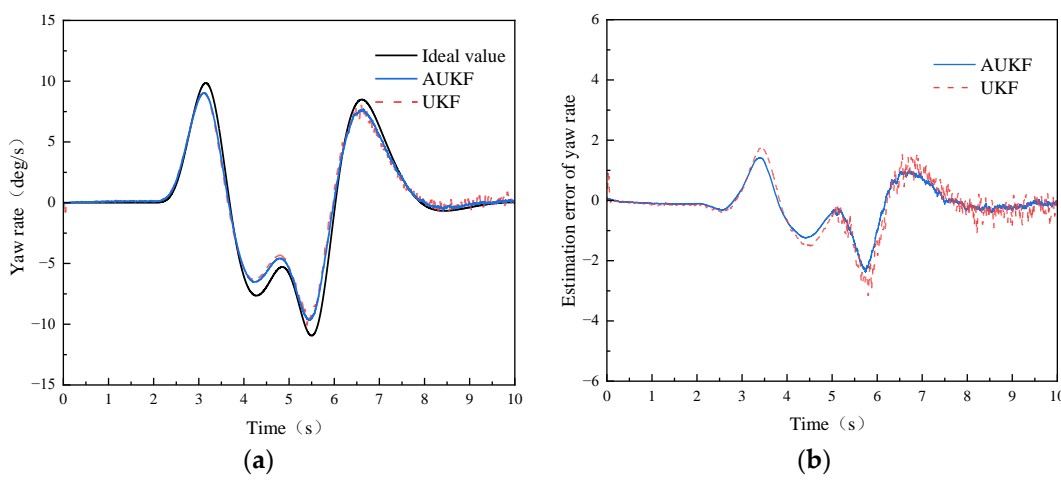

**(a)**  **(b)**

**Figure 6.** Simulation results of yaw rate. (**a**) Comparison of estimated results and (**b**) comparison of estimated error.

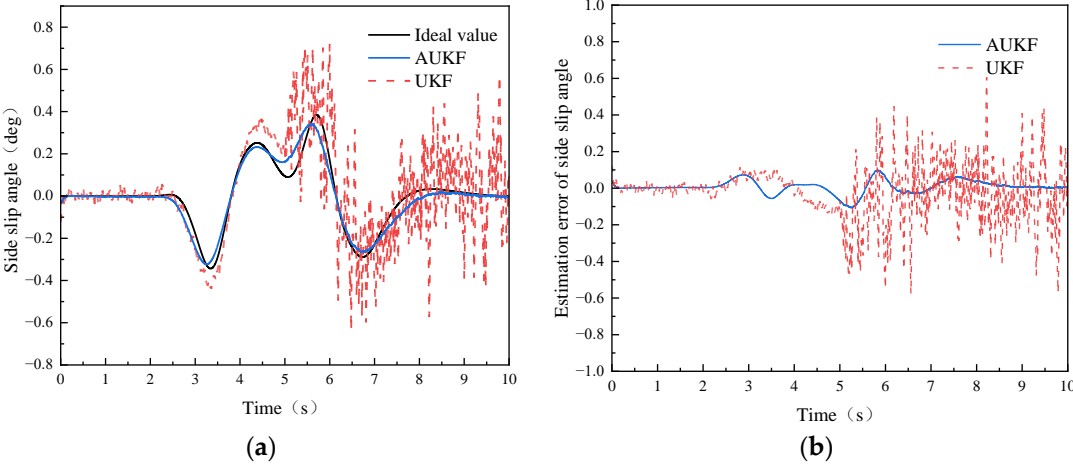

**Figure 7.** Simulation results of side slip angle. (**a**) Comparison of estimated results and (**b**) comparison of estimated error.

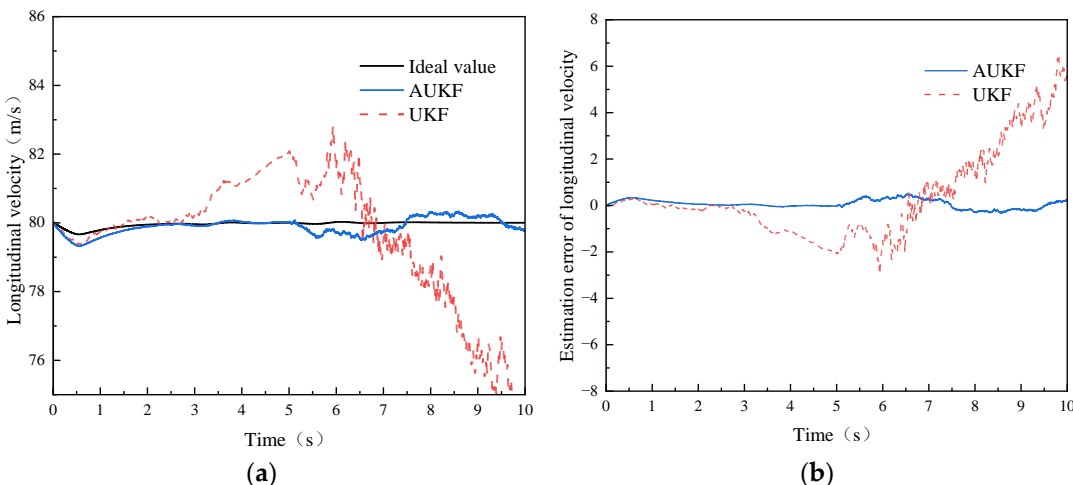

**Figure 8.** Simulation results of longitudinal velocity. (**a**) Comparison of estimated results and (**b**) comparison of estimated error.

In Figures 6a and 7a, we can see that the vehicle changes lanes at 5–15 s. At the inflection point of the curve, the standard UKF deviates from the reference value, while the AUKF can track the ideal value effectively. At the same time, due to the change in process noise after 10 s, UKF diverges in the estimation, which is larger than that in the previous 10 s, indicating that UKF algorithm cannot accurately estimate the corresponding state parameters when there are different degrees of noise disturbance. The estimation results of the AUKF algorithm are largely consistent with the ideal values in the whole working condition, and the effect of the AUKF algorithm is ideal. The maximum instantaneous error of the yaw rate and side slip angle is 2.358 deg/s and 0.104 deg.

As shown in Figure 8a, the UKF algorithm has a divergence in the estimation, especially after the variance of the noise matrix increases tenfold, which shows that the UKF algorithm cannot accurately estimate the corresponding state parameters when there are different degrees of noise disturbance. The AUKF in the first 10 s is largely consistent with the ideal value, and the error is within an acceptable range in the last 10 s. The effect of the AUKF algorithm is ideal and the maximum instantaneous error of the longitudinal velocity is 0.495 m/s.

The estimation errors of each algorithm are shown in Figures 6b, 7b and 8b. The RMSE of the UKF and AUKF are given in Table 2.

**Table 2.** RMSE of the two algorithms during the entire duration of the process.

| Estimation Algorithm | RMSE of State VARIABLE Estimation | | |
|---|---|---|---|
| | Yaw Rate | Side Slip Angle | Longitudinal Velocity |
| UKF | 0.8539 | 0.1426 | 1.9616 |
| AUKF | 0.6824 | 0.0363 | 0.1980 |

As can be seen from Table 1, in the estimation of the three kinds of vehicle states, the RMSE estimated by the AUKF algorithm is the smallest, demonstrating its higher estimation accuracy. Compared with the UKF algorithm, the estimation accuracy for yaw rate, side slip angle and longitudinal velocity state variables was improved by 20.08%, 74.54% and 89.91%, respectively.

In summary, compared with the UKF algorithm, the AUKF algorithm can better suppress the interference of noise, demonstrating its higher estimation accuracy and stronger robustness.

### 4.2. Simulation Analysis of Serpentine Condition

(1) We fixed the vehicle speed at 40 km/h and initialize the state vector as $x_0 = [0,0,40/3.6]$. Figures 9a, 10a and 11a show the simulation results of UKF, AUKF and ideal values.

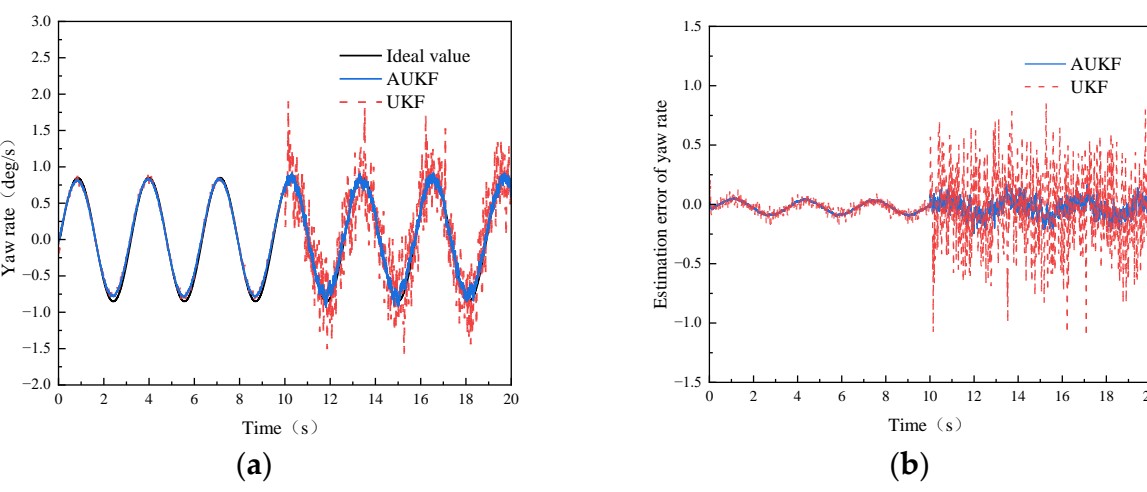

**(a)**       **(b)**

**Figure 9.** Simulation results of yaw rate. (**a**) Comparison of estimated results and (**b**) comparison of estimated error.

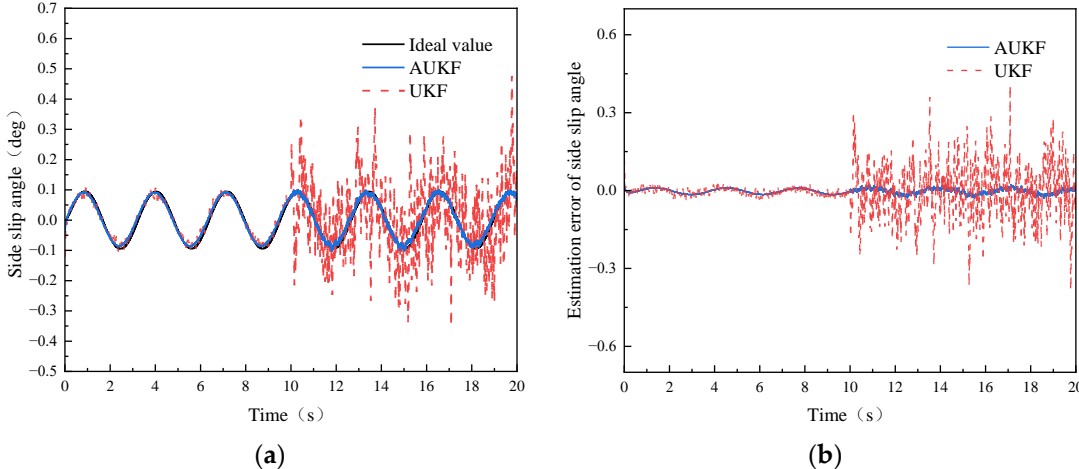

**(a)**       **(b)**

**Figure 10.** Simulation results of side slip angle. (**a**) Comparison of estimated results and (**b**) comparison of estimated error.

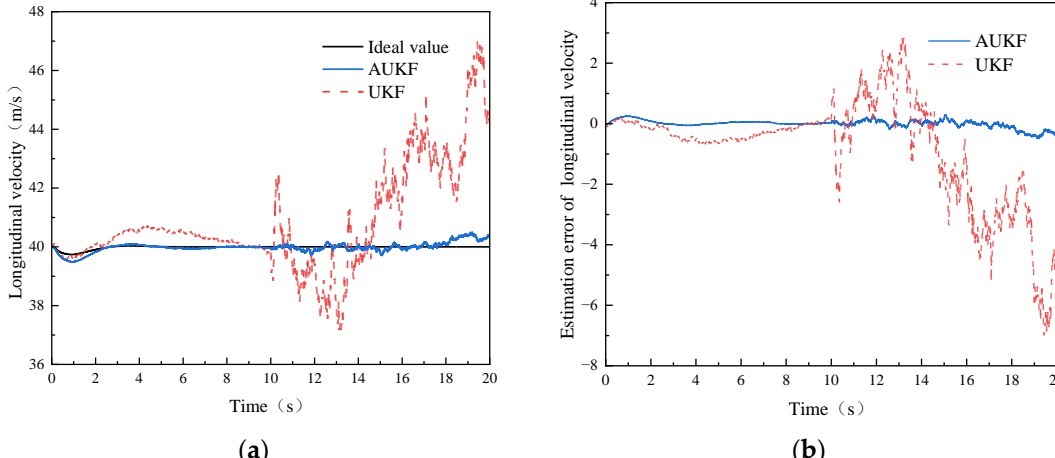

**Figure 11.** Simulation results of longitudinal velocity. (**a**) Comparison of estimated results and (**b**) comparison of estimated error.

In Figure 9a, we can see that the standard UKF estimates a divergence at the inflection point of the curve during 0–10 s, while the AUKF can track the ideal value effectively. At the same time, due to the change in process noise after 10 s, the divergence phenomenon of the UKF is more obvious in the estimation, and this fluctuation is larger than that seen during the first 10 s, which shows that the UKF algorithm cannot accurately estimate the corresponding state parameters when there are different degrees of noise disturbance. However, the estimation results of the AUKF algorithm are largely consistent with the ideal values in the whole working condition, and the effect of the AUKF algorithm is ideal. The maximum instantaneous error of the yaw rate is 0.247 deg/s.

In Figure 10a, we can see that the standard UKF estimates a divergence at the inflection point of the curve during 0–10 s, while the AUKF can track the ideal value effectively. At the same time, due to the change in process noise after 10 s, the divergence phenomenon of UKF is more obvious in the estimation, and this fluctuation is larger than that seen in the first 10 s, which shows that the UKF algorithm cannot accurately estimate the corresponding state parameters when there are different degrees of noise disturbance. However, the estimation results of the AUKF algorithm are largely consistent with the ideal values in the whole working condition, and the effect of the AUKF algorithm is ideal. The maximum instantaneous error of the side slip angle is 0.038 deg.

As shown in Figure 11a, when the UKF algorithm diverges in estimation, especially after the variance of the noise matrix increased tenfold, the divergence is particularly obvious, indicating that the UKF algorithm cannot accurately estimate the corresponding state when there are different degrees of noise disturbance. However, the estimated result of the AUKF algorithm in the first 10 s is largely consistent with the ideal value, and the error between the estimated result and the reference value in the next 10 s is within an acceptable range, and the effect of the AUKF algorithm is ideal. The maximum instantaneous error of the longitudinal velocity is 0.492 m/ s.

The estimation errors of each algorithm are shown in Figures 9b, 10b and 11b. The RMSE of the UKF and AUKF are given in Table 3.

**Table 3.** RMSE of the two algorithms during the entire duration of the process.

| Estimation Algorithm | RMSE of State Variable Estimation | | |
|:---:|:---:|:---:|:---:|
| | Yaw Rate | Side Slip Angle | Longitudinal Velocity |
| UKF | 0.2074 | 0.0730 | 1.9674 |
| AUKF | 0.0631 | 0.0101 | 0.1314 |

As can be seen from Table 3, in the estimation of the three kinds of vehicle states, the RMSE estimated by the AUKF algorithm is the smallest, demonstrating its higher estimation accuracy. Compared with UKF algorithm, the estimation accuracy of yaw rate, side slip angle and longitudinal velocity state variables was improved by 69.58%, 86.16% and 93.32%, respectively.

(2) We fixed the vehicle speed at 80 km/h and initialized the state vector as $x_0 = [0,0,80/3.6]$. Figures 12a, 13a and 14a show the simulation results of UKF, AUKF and ideal values.

In Figure 12a, we can see that the standard UKF estimates a divergence at the inflection point of the curve during 0–10 s, while the AUKF can track the ideal value effectively. At the same time, due to the change in process noise after 10 s, the divergence phenomenon of the UKF is more obvious in the estimation, and this fluctuation is larger than that seen during the first 10 s, which shows that the UKF algorithm cannot accurately estimate the corresponding state parameters when there are different degrees of noise disturbance. However, the estimation results of the AUKF algorithm are largely consistent with the ideal values in the whole working condition, and the effect of the AUKF algorithm is ideal. The maximum instantaneous error of the yaw rate is 0.617 deg/s.

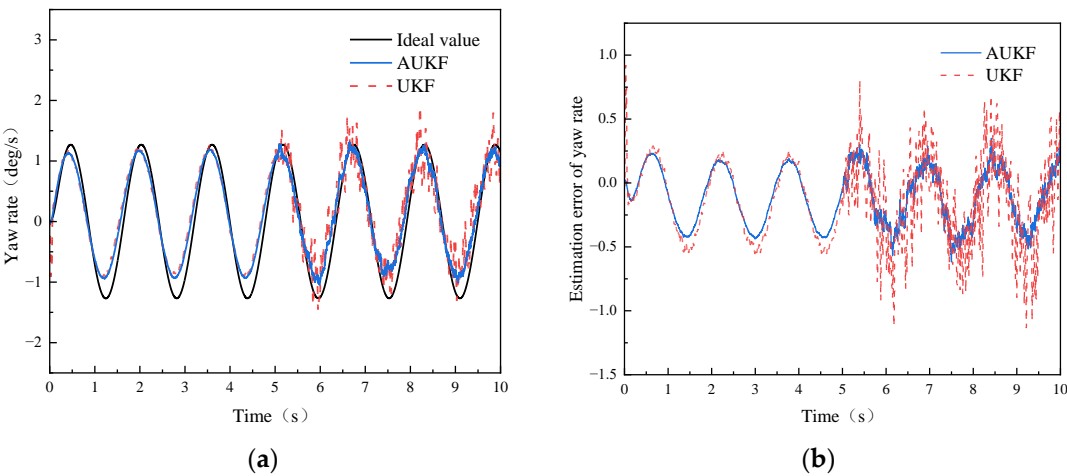

**Figure 12.** Simulation results of yaw rate. (**a**) Comparison of estimated results and (**b**) comparison of estimated error.

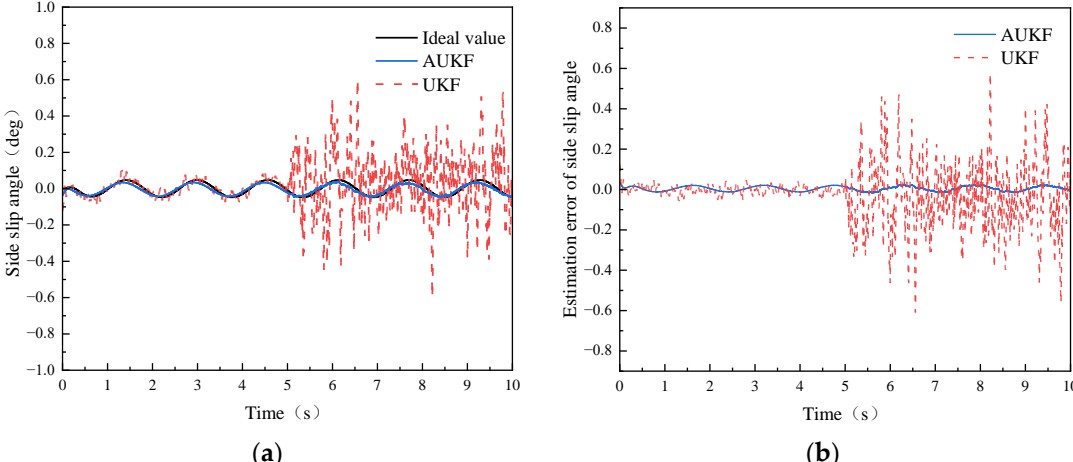

**Figure 13.** Simulation results of side slip angle. (**a**) Comparison of estimated results and (**b**) comparison of estimated error.

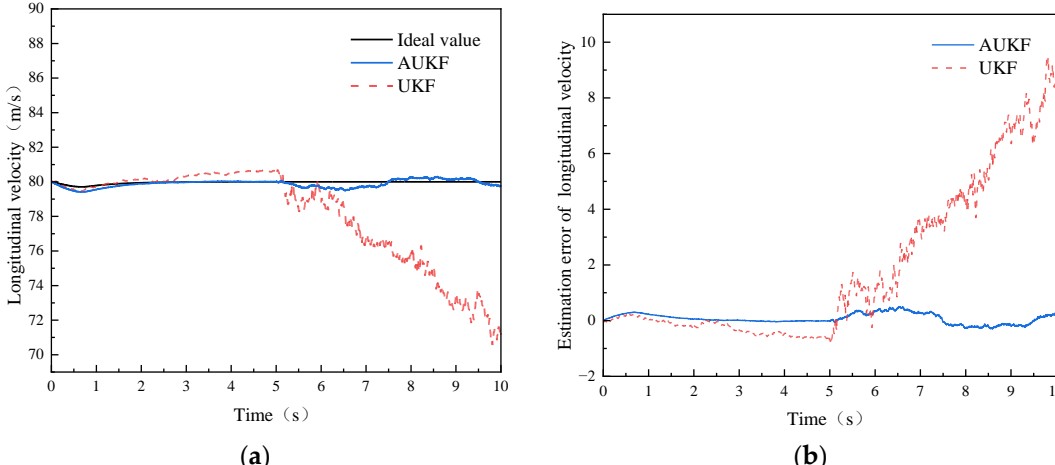

**Figure 14.** Simulation results of longitudinal velocity. (**a**) Comparison of estimated results and (**b**) comparison of estimated error.

In Figure 13a, we can see that the standard UKF estimates a divergence at the inflection point of the curve during 0–10 s, while the AUKF can track the ideal value effectively. At the same time, due to the change in process noise after 10 s, the divergence phenomenon of UKF is more obvious in the estimation, and this fluctuation is larger than that seen in the first 10 s, which shows that the UKF algorithm cannot accurately estimate the corresponding state parameters when there are different degrees of noise disturbance. However, the estimation results of the AUKF algorithm are largely consistent with the ideal values in the whole working condition, and the effect of the AUKF algorithm is ideal. The maximum instantaneous error of the side slip angle is 0.038 deg.

As shown in Figure 14a, when the UKF algorithm diverges in estimation, especially after the variance of the noise matrix increases tenfold, the divergence is particularly obvious, indicating that the UKF algorithm cannot accurately estimate the corresponding state when there are different degrees of noise disturbance. However, the estimated result of the AUKF algorithm in the first 10 s is largely consistent with the ideal value, and the error in the next 10 s is within an acceptable range. The effect of the AUKF algorithm is ideal. The maximum instantaneous error of the longitudinal velocity is 0.505 m/s.

The estimation errors of each algorithm are shown in Figures 12b, 13b and 14b. The RMSE of the UKF and AUKF are given in Table 4.

**Table 4.** RMSE of the two algorithms during the entire duration of the process.

| Estimation Algorithm | RMSE of State Variable Estimation | | |
| :---: | :---: | :---: | :---: |
| | Yaw Rate | Side Slip Angle | Longitudinal Velocity |
| UKF | 0.3465 | 0.1260 | 3.3560 |
| AUKF | 0.2476 | 0.0125 | 0.1942 |

As can be seen from Table 4, in the estimation of the three kinds of vehicle states, the RMSE estimated by the AUKF algorithm is the smallest, demonstrating its higher estimation accuracy. Compared with UKF algorithm, the estimation accuracy of yaw rate, side slip angle and longitudinal velocity state variables was improved by 28.54%, 90.08% and 94.21%, respectively.

In summary, compared with the UKF algorithm, the AUKF algorithm can better suppress the interference of noise, demonstrating higher estimation accuracy and stronger robustness.

## 5. Conclusions

When the vehicle is disturbed by different degrees of noise during driving, the traditional vehicle state estimation methods will show some problems such as a divergence or

even a failure of the estimation results, which will affect the decision making and control of subsequent vehicle systems. On this basis, we propose a MAP-based AUKF algorithm to solve the problem of adaptive estimation of vehicle state parameters under different degrees of noise interference. In this study, the maximum a posteriori algorithm was used to dynamically update the noise of a vehicle system, and it was embedded into the update steps of an UKF to form an AUKF. Through the simulation experiments under double lane change and serpentine conditions, our method can adapt to different levels of noise interference and obtain great estimation accuracy. The estimation accuracy for the yaw rate, side slip angle and longitudinal velocity was improved by 20.08%, 40.98% and 89.91%, respectively. Because the AUKF has a better performance, this method is expected to provide more reliable perceptual information for intelligent driving vehicle decisions and control system applications.

The next steps include building a more accurate vehicle model and taking into account the roll motion of the vehicle, the motion of the suspension and the nonlinear characteristics of the tire. The proposed algorithm could also be tested with real vehicles.

**Author Contributions:** Conceptualization, Y.W.; Writing—original draft, Y.W.; Writing—review & editing, Y.L. and Z.Z.; Supervision, Y.L. and Z.Z. All authors have read and agreed to the published version of the manuscript.

**Funding:** This research was partially funded by Jilin Provincial Major Science and Technology Projects (Grant: 20210301020GX).

**Data Availability Statement:** Data were curated by the authors and are available upon request.

**Conflicts of Interest:** The authors declare no conflict of interest.

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
