# Peer review of "State Parameter Estimation of Intelligent Vehicles Based on an Adaptive Unscented Kalman Filter"

_electronics, doi:10.3390/electronics12061500_

Round 1

Reviewer 1 Report

The paper is interesting, but some changes and additions need to be made before it can be published.

I. Regarding the 3 DOF vehicle model:

1. Indeed the center of gravity is considered in the longitudinal plane of symmetry of the vehicle (X axis). But I don't understand why in Figure 1 the center of gravity and the longitudinal plane of symmetry of the vehicle is at the wheels.

2. Why is Vx2 facing outwards?

3. What do the parameters: v and u represent?

4. The representation of the velocity at the center of gravity should be represented in the plane of the front wheels, which are turned.

5. What does r represent? In which plane does r act? Should in the XOY plane.

6. All parameters in Figure 1 should be explained.

7. Figure 1 should be detailed and corrected.

II. Other remarks:

In Proposition: ”2. Calculated Rk by equation (19); if R<0, then" it is an error.  Should be Qk, not Rk.

"i=L+1,...2L"  I don't think it is correct.  After to relation (8) it is written to the exponent: 1,...,2L.

In relations (7), (10), (11) and others it is written: the sum from i=0 to 2L. Is i=0 correct?

All graphs should be discussed much more.

Finally, I reiterate that it is important for the authors to verify the correctness of the vehicle model and to describe it in detail in the paper.

Author Response

请参阅附件。

Reviewer 2 Report

1. Authors contribution is not clearly given, and the novelty of the work is not clearly presented in the work.

2. Reference work is very weak, need to improve a lot with recent related works 

3. section 3 is the existing algorithm why the authors have given separate section for these mathematical equations.

Reviewer 3 Report

This article proposes "State Parameter Estimation of Intelligent Vehicles Based on Adaptive Unscented Kalman Filter." My comments are listed below.

1. Noise adaptive unscented Kalman filter has been proposed in some others articles. The novelty of the proposed filter should be elaborated in the abstract section. Is the novelty lies on the application in State Parameter Estimation of Intelligent Vehicles? Also, please add some numerical results in the abstract section. 

2. The second last paragraph of the introduction section has two consecutive "on the other hand". please correct it. In the third paragraph of the introduction section unscented Kalman filter is discussed. However, the gaps have not been properly defined which are to be filled by this research. 

3. There is no citations from the year 2021 and 2022. Thus, the introduction section needs to be carefully improved with recent studies to define the research gaps. 

4. The main contribution of this work is described in section 3. However, it introduced equations from the existing literature without proper citations and explanations of different terms. 

5. Please describe the impact of forgetting factor (b) on the performance of the proposed algorithms in the results section. 

6. Yaw rate, and Side slip angle improvement with the proposed  work is not sufficient as contributions. 

Overall, this work needs a lot of improvements for further considerations. 

Round 2

Reviewer 1 Report

The authors verified and proved that the model is correct. They answered all my comments. However, there would be one aspect to clarify:

Why in Figure 2 is the index "k+1", and in equations (6), (8), (14) and (15) is the index "k-1" or "k". Correct ?

Equation 22 must be written in the same font as the other equations.

Reviewer 2 Report

The authors advised checking the grammar mistakes and explaining the inference got from the results.

Reviewer 3 Report

The authors have addressed all comments. However, I feel the number of references should be at least 30. Thus, a minor revision is needed to improve the introduction section. 
